

# Visual analysis of model parameter sensitivities along warm conveyor belt trajectories using Met.3D (1.6.0-multivar0)

Christoph Neuhauser[1], Maicon Hieronymus[2], Michael Kern[3], Marc Rautenhaus[4], Annika Oertel[5], and Rüdiger Westermann[1]

[1]School of Computation, Information and Technology, Technical University of Munich, Garching bei München, Germany
[2]Institute of Computer Science, Johannes Gutenberg University, Mainz, Germany
[3]Advanced Micro Devices GmbH, Dornach bei München, Germany
[4]Regional Computing Centre, Visual Data Analysis Group, University of Hamburg, Hamburg, Germany
[5]Institute of Meteorology and Climate Research, Karlsruhe Institute of Technology, Karlsruhe, Germany

**Correspondence:** Christoph Neuhauser (christoph.neuhauser@tum.de)

**Abstract.** Numerical weather prediction models rely on parameterizations for subgrid-scale processes, e.g., for cloud microphysics, which are a well-known source of uncertainty in weather forecasts. Via algorithmic differentiation, which computes the sensitivities of prognostic variables to changes in model parameters, these uncertainties can be quantified. In this article, we present visual analytics solutions to analyze interactively the sensitivities of a selected prognostic variable to multiple model parameters along strongly ascending trajectories, so-called warm conveyor belt (WCB) trajectories. We propose a visual interface that enables to a) compare the values of multiple sensitivities at a single time step on multiple trajectories, b) assess the spatio-temporal relationships between sensitivities and the trajectories' shapes and locations, and c) find similarities in the temporal development of sensitivities along multiple trajectories. We demonstrate how our approach enables atmospheric scientists to interactively analyze the uncertainty in the microphysical parameterizations, and along the trajectories, with respect to the selected prognostic variable. We apply our approach to the analysis of WCB trajectories within the extratropical cyclone "Vladiana", which occurred between 22-25 September 2016 over the North Atlantic.

## 1 Introduction

The warm conveyor belt (WCB) is a well-defined moist airstream, which originates in the lowermost levels of the atmosphere within an extratropical cyclone's warm sector and generally ascends poleward to the upper troposphere within two days (Wernli, 1997; Madonna et al., 2014). WCBs play a critical role in cloud formation and precipitation in the extratropics (e.g., Madonna et al., 2014; Pfahl et al., 2014). In data from numerical weather prediction (NWP) models, WCBs are often detected and analyzed by means of trajectories computed from the simulated time-dependent 3-D wind fields (e.g., Wernli, 1997; Rautenhaus et al., 2015a). Coherent ensembles of trajectories are then used to analyze processes not directly discernible from the underlying wind fields, including the origins of moist airflow and how precipitation patterns emerge from airmass ascent.

Surface precipitation rates in extratropical cyclones can be significantly impacted by convective ascent embedded in WCBs (Oertel et al., 2020, 2021; Jeyaratnam et al., 2020). Moreover, the precipitation formation pathway and associated latent heating



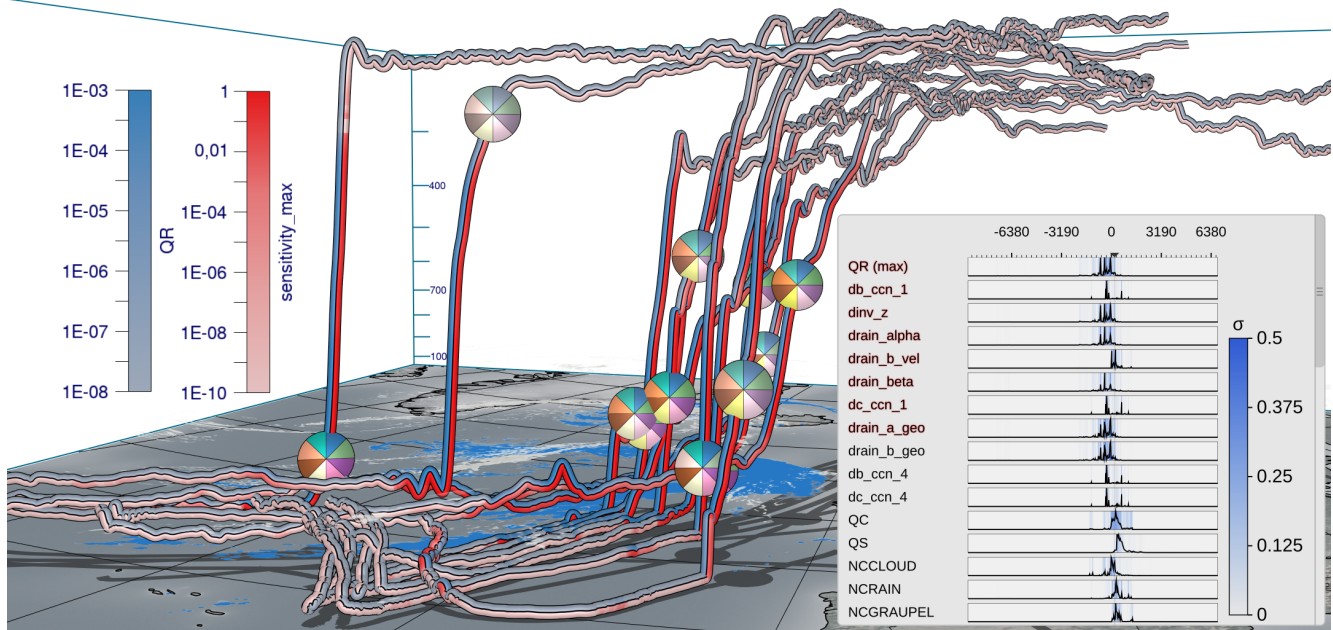

**Figure 1.** Visual analysis of the sensitivity of a prognostic variable to selected model parameters (emphasized in red in curve plot overlay) along warm conveyor belt trajectories in the extratropical cyclone "Vladiana", to assess uncertainties of parameterizations in numerical weather prediction models. Prognostic variable (blue) and maximum sensitivity (red) are color coded along the trajectories in view-aligned bands, so that one half of the seen trajectory tube is consumed by either color. Multiple sensitivities at a selected time step are visualized via pie charts that are mapped onto spheres in the 3-D view. A consistent view-aligned mapping of sensitivities to pie charts enables an effective comparison across the trajectories. A curve plot shows statistical summaries of prognostic variables, sensitivities, and model parameters to which sensitivities are computed. Surface precipitation is shown on the ground in blue. Display of the earth's surface and shadows place trajectories in spatial context.

are sensitive to the cloud microphysical processes implemented in the numerical model, and may in turn, introduce uncertainties to WCB ascent (Joos and Forbes, 2016; Mazoyer et al., 2021). As the scale of cloud microphysical processes responsible for precipitation formation is too small to be explicitly resolved in NWP models, parameterizations are used to calculate the

25 integrated effects on the resolved prognostic variables. These parameterization schemes, however, are still associated with large uncertainties that can influence the representation of atmospheric dynamics including airmass ascent and formation of precipitation in NWP models (Leutbecher and Palmer, 2008; Ollinaho et al., 2017; Pickl et al., 2022).

Thorough analysis of the impact of the parameterizations' parameters on prognostic variables can clarify how, when, and where model representations of atmospheric processes including airmass ascent and formation of precipitation are particularly

sensitive, and can yield enhanced process understanding and eventually improved parameterizations. Such analysis has motivated our work. On the one hand, it requires a methodology to efficiently compute the sensitivities, on the other hand it requires an approach to locate sensitive behaviour in space and time and to place it into the context of the simulated atmospheric pro-





cesses. Regarding the efficient computation of sensitivities, we follow up on recent work by Hieronymus et al. (2022) using Algorithmic Differentiation (AD), a method to compute derivatives of an implemented model (Griewank and Walther, 2008).

In this article, we present a novel method for the visual analysis of sensitive behaviour in space and time. We propose an interactive visualization workflow to facilitate

- automatic identification of relevant sensitivities,

- simultaneous visualization of multiple sensitivities,

- and linking of sensitivities to trajectories in 3-D space.

We note that while the visualization method we are presenting has been motivated by the analysis of sensitivities of WCB trajectories, it can readily also be applied to further analysis of trajectory data that requires the simultaneous display and analysis of multiple variables.

Visualization approaches for meteorological analysis have been discussed widely in the literature. Comprehensive overviews have been provided by Rautenhaus et al. (2018); Afzal et al. (2019); Yoshizumi et al. (2020). Our workflow builds upon and extends approaches to perform interactive statistical data analysis (Love et al., 2005; Potter et al., 2010; Orf et al., 2016; Meyer et al., 2021), and touches on aspects of 3-D feature-based visualization (Rautenhaus et al., 2015a; Kern et al., 2018, 2019; Bader et al., 2019; Kappe et al., 2022; Bösiger et al., 2022). While in the current work AD is used to compute uncertainty information in the form of parameter sensitivities, previous works in visualization have mostly addressed simulation uncertainty in the form of given simulation ensembles (Sanyal et al., 2010; Wang et al., 2018; Rautenhaus et al., 2018).

In this study, we discuss the use of the proposed workflow for analysis of WCB trajectories associated with the extratropical cyclone "Vladiana", which occurred between 22-25 September 2016 over the North Atlantic (Schäfler et al., 2018). The following analysis questions motivated our work:

1. Do similar trends regarding selected sensitivities and prognostic variables occur across a (sub-)group of selected trajectories? (Q1)

2. Do different sensitivities and prognostic variables show similar statistical characteristics across a selected trajectory group? (Q2)

3. How do sensitivities depend on the time and location along the trajectories, and how are they related to, e.g., precipitation and cloud formation? (Q3)

4. Do coherent sensitivity patterns emerge if trajectories ascending at different times are considered relative to their time of ascent? (Q4)

5. Do sensitivities differ with respect to different types of trajectories (i.e., convective vs. slantwise)? (Q5)

While Q1 to Q3 enable improved process understanding, Q4 and Q5 provide insight into the structure of WCB trajectories and their associated sensitivities. Figure 1 provides a typical visualization of our workflow, which combines standard and novel



visualization techniques. For an impression of the interactive aspects, we refer to the Supplementary Videos 1 and 2, which
provide an overview of the implemented visualization techniques (Video 1) and illustrate the analysis of the "Vladiana" WCB
trajectories (Video 2).

The article is structured as follows. We first introduce the employed data and the method's interactive workflow (Sect. 2),
before the proposed visualization techniques (Sect. 3) and their technical implementation (Sect. 4) are discussed in detail. In
Sect. 5, the visualization techniques are applied to WCB trajectories to illustrate the sensitivity of rain mass density to various
microphysical parameters. Section 6 concludes with a summary.

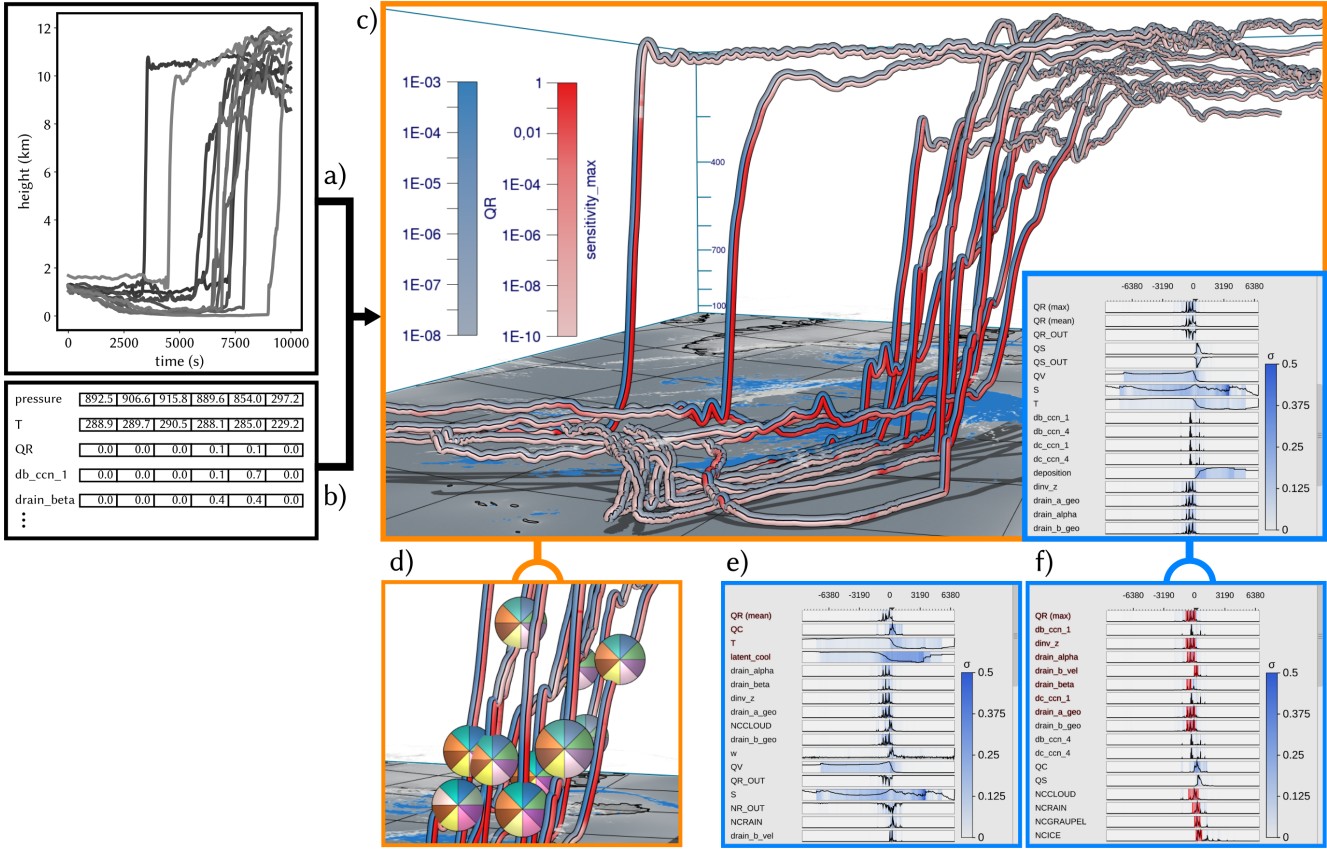

**Figure 2.** Workflow overview: Met.3D reads a) 3-D trajectory data and b) tables of model variables and sensitivities along the trajectories. c)
The visualization canvas of Met.3D, including the 3-D trajectory view that is linked to the curve plots summary view. d) Focus view options
using sphere-based multi-parameter visualization via pie charts. e) Statistical summaries of the temporal development of variables and
sensitivities, which can be ordered automatically regarding the similarity of their temporal development to a selected variable or sensitivity.
f) Variables exhibiting a selected sequence of events can be determined automatically and shown first.





## 2 Data and method overview

The proposed workflow and methodology facilitates the interactive visual analysis of the effects of simulation model parameters on a selected target variable. In this study, we focus on rain mass density along convective warm conveyor belt trajectories, which are responsible for heavy rainfall on the earth's surface. The analysis hints on relationships between the trajectories' spatial locations and shapes, and the occurrence of specific features in the sensitivities of the selected variable to different model parameters.

### 2.1 Data

We consider WCB trajectories that are computed for the extratropical cyclone "Vladiana", which developed from 22-25 Sep 2016 in the North Atlantic during the North Atlantic Waveguide and Downstream Impact Experiment field campaign (Schäfler et al., 2018). The trajectory data of the case-study shown here is taken from a simulation described in detail by Oertel et al. (2020) with the NWP model COSMO version 5.1 (Baldauf et al., 2011). In addition, an online trajectory scheme (Miltenberger et al., 2013) was applied to calculate the positions and properties of the trajectories from the resolved 3-D wind field at every model time step, here 20 s.

In this work, sensitivity is defined as the linearly predicted change of a prognostic variable if a model parameter is perturbed by 10 % (Hieronymus et al., 2022). The prognostic variable can be any of the NWP simulation output. In this work, we focus on multi-parameter sensitivities of rain mass density (QR). The linear prediction is the gradient computed via AD times 10 % of the model parameter value. AD can be used to quantify the impact of multiple model parameters on a prognostic variable at once. It exploits the fact that any computer model after code compilation becomes a sequence of differentiable elemental operations. By repeatedly applying the chain rule, the derivative for any code can be calculated automatically alongside the usual run of the code. AD has been applied on a warm-rain microphysics scheme for idealized trajectories (Baumgartner et al., 2019), and recently on convective and slantwise WCB trajectories (Hieronymus et al., 2022). The application of AD to a prognostic variable along WCB trajectories results in one sensitivity value of this variable for each model parameter and on each simulation point along the trajectories. In an NWP model with multiple processes and hundreds of parameters, AD also reveals which processes are active. That is, if the sensitivity to a parameter is above zero, then the simulation must have involved the corresponding process.

AD has been applied to convective and slantwise trajectories in "Vladiana" with the tool by Hieronymus et al. (2022), which implements the Seifert and Beheng (2006) two moment cloud microphysics model. The tool includes routines for the ice phase (Kärcher et al., 2006; Phillips et al., 2008) and is augmented with CoDiPack (Sagebaum et al., 2019) to evaluate the Jacobian of the implemented model at every time step in an efficient way. Overall, the sensitivities of rain mass density with respect to 177 model parameters have been computed via AD, of which the 40 most important parameters are used in this work. For an overview over all available parameters and prognostic variables, please refer to Appendix C.





## 2.2 Method overview

Figure 2 shows an overview of the method's workflow. The input is a set of convective WCB trajectories which have been computed over a time interval of interest, and a set of model parameter sensitivities along these trajectories with respect to a selected prognostic variable (Fig. 2b). Sensitivities are named "d[...]", which stands for $\partial QR/\partial[...]$, where rain mass density (QR, kg m$^{-3}$) is the selected target variable, and "[...]" is the model parameter in question. "sensitivity_max" is the per-time maximum of all sensitivities.

As shown in Figure 1 and Figure 2e, we use an interactive multi-parameter "curve plot" (2-D line plot) to enable the user to analyze the time evolution of the maximum of any selected sensitivity (as well as the standard deviation (stdev) to this maximum) over all trajectories. Beyond this, sensitivities can be sorted automatically with respect to their temporal development, by using the development of a selected sensitivity as reference. The user can then select a time period in the curve plot of a sensitivity or prognostic variable and let the system search for similar trends in the temporal developments of other sensitivities or prognostic variables. The curve plot view enables an interactive comparative visualization of the statistical similarities of local and global temporal trends across the set of selected trajectories.

Curve plots alone, however, cannot reveal the relationships between sensitivities and the trajectories' locations and shapes. Therefore, the curve plots are embedded into the open-source meteorological 3-D visualization system Met.3D (Rautenhaus et al., 2015b). Met.3D visualizes the trajectories in their spatial context (i.e., the 3-D trajectory view), including visualizations of additional data sources like textured terrain fields, and in particular 3-D atmospheric field data. From its existing support to display a single parameter along 3-D trajectories (Rautenhaus et al., 2015a), Met.3D has been extended according to the specific visualization options required to support a comparative analysis as mentioned. Multiple sensitivities along a trajectory can now be shown via stripe patterns with different colors (Neuhauser et al., 2022), and by using additional geometric primitives like enlarged disks (Sadlo et al., 2006).

The curve plot view is linked to the trajectory view in that the user can move a vertical line along the time axis, and instantly the points on each trajectory corresponding to that time are highlighted by enlarged disks (sphere glyphs), which encode multiple sensitivities simultaneously and enable a comparison of sensitivities across trajectories (cf. Video 1, 02:46 min). Alternatively to moving the time line in the curve plot, the user can pick a sphere glyph and move it along the trajectory (cf. Video 1, 03:18 min). All other glyphs are moved accordingly in time so that via animation the sensitivities on different trajectories can be compared.

Striped bands become problematic when the bands are fixed to a bending trajectory's surface, where they appear distorted and can cover differently sized regions in the view plane (see Fig. 6a). Similarly, enlarged disks suffer from occlusions under certain views, and disks may penetrate each other when the trajectory exhibits strong bending. To address these limitations, we use view-aligned bands (Russig et al., 2023) that consistently segment the visible surface part into equally sized and connected stripes. By further letting the system automatically compute for each trajectory its unique time of ascent and interpreting the current time relative to these times, the trajectories' sensitivities during the ascend phases can be effectively compared.



As soon as more than two to three sensitivities are visualized simultaneously, however, the single bands become too thin and can hardly be distinguished in the 3-D view. Therefore, we restrict to showing only the temporal evolution of the target variable and the maximum sensitivity over all parameters via colored bands, and propose a view-aligned circular mapping for showing simultaneously multiple sensitivities at a selected time step. Multiple sensitivities are encoded via a pie chart that has a fixed orientation in view-space and is mapped onto a sphere centered at a trajectory point. The enlarged sphere acts both as

a time step marker and magnifying lens. Since pie-charts on different trajectories are consistently oriented in view-space, the sensitivities can be compared effectively in a single view. The number of subdivisions of the pie chart is given by the number of sensitivities the user selects in the curve plot view.

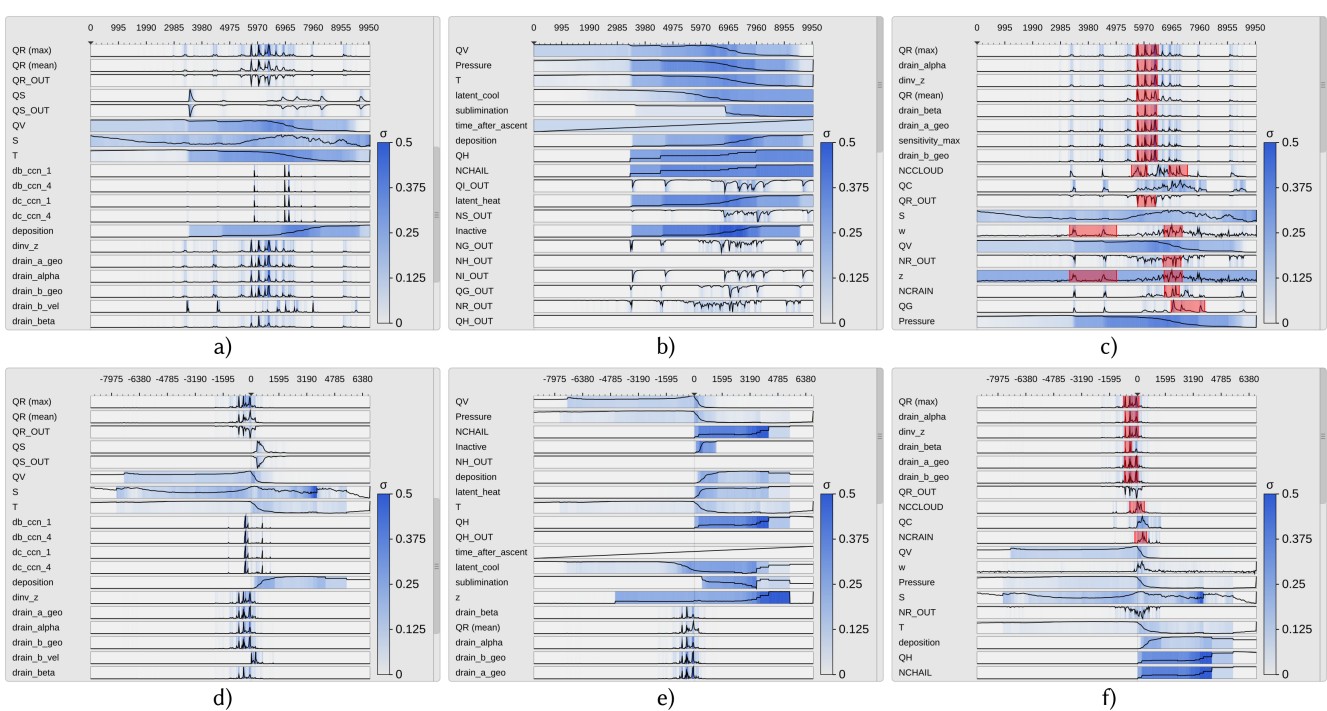

**Figure 3.** Curve plots with trajectories aligned by time step (a,b,c) and time of ascent (d,e,f). Curve plots show the mean of prognostic variables, and the maximum of sensitivities over all trajectories. The stdev to the mean/maximum is mapped to color. Top curve shows target variable rain mass density (QR). a,d) Curve plots in random order. b,e) Curve plots are sorted regarding the similarity of their time development relative to the target variable QV (cloud mass density). c,f) Sorting regarding similarity to max QR. A pattern of consecutive spikes has been selected in QR, and regions in which similar features have been determined are highlighted.





## 3 Visualization techniques

The visual analysis workflow presented in this work builds upon the curve plot view, the 3-D trajectory view, and interac-
tive linkage between these two views. Linkage enables to find relationships between locations with high sensitivities along
trajectories and the trajectories' locations and shapes.

### 3.1 Multi-parameter curve plot view

The curve plot view shows the single curve plots of the prognostic variables and sensitivities vertically aligned (cf. Fig. 3). The
time axis is going to the right and the vertical axis represents the value domain. All values are initially normalized to $[0, 1]$. The
trajectories are traced with a time step of $\Delta t = 20\,s$, which is also the time delta between two data points in the horizontal axis.
When the number of time steps exceeds the number of pixels reserved for showing the curve plots, the algorithm *largest triangle
three buckets* (LTTB) (Steinarsson, 2013) is used to recursively downsample the data. LTTB takes into account the perceptual
importance of points during the downsampling process by assessing the area of triangles formed by points in neighboring
buckets. In this way, the performance penalty of drawing too many points can be avoided, simultaneously ensuring that no
features are lost. By generating the curve plots at multiple resolutions, the user can zoom into interesting time intervals and
analyze the variables and sensitivities over these intervals in more detail.

For the target variable and sensitivities, in each band the maximum over all trajectories is shown via a curve. For all other
prognostic variables and model parameters the mean over all trajectories is shown. Since the sensitivities are often close to
zero, resulting in very small mean values, the maximum values and corresponding stdevs can far more effectively indicate the
spread of the distributions and the overall trend regarding their strengths. In particular, regions of potential local instability are
emphasized and high sensitivities are not missed. The background is colored according to the stdevs with respect to the values
represented by the curves, i.e., stdev is mapped to a color ranging from white (low value) to blue (high value). By utilizing
mouse controls, the user can scroll through the set of parameters and zoom into individual regions in the curve plot view. A
moveable vertical line indicates the currently selected time step.

Since there are many parameters and not all can be shown in one single view, the system proposes an automatic ordering
to quickly identify sets of parameters with similar sensitivity development over time. Therefore, the user selects an individual
curve plot, and the system sorts all curve plots in descending order regarding the similarity to the curve in this plot. As a
measure of similarity we use the absolute normalized cross-correlation

$$NCC(X,Y) = \frac{1}{N}\sum_i \frac{(X_i - \mu_x)(Y_i - \mu_y)}{\sigma_x \sigma_y}. \tag{1}$$

Here, $X_i$ and $Y_i$ are two time series, and $\mu_x, \mu_y$ and $\sigma_x, \sigma_y$ the corresponding means and stdevs. Note that due to the division
by the stdev, NCC becomes independent of the scale of the two time series.

We further considered CrossMatch (Toyoda and Sakurai, 2013) and the "edit distance on real sequence" (EDR) (Chen et al.,
2005) as alternatives for similarity sorting. However, since the former does not support data normalization, and the latter may
suppress relevant sensitivities due to built-in noise suppression, both turned out to be less effective in our scenario.





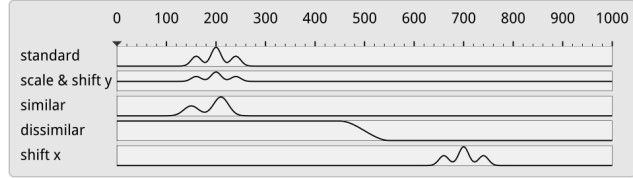

**Figure 4.** Test sequences sorted by their similarity to "standard" using the absolute NCC. The NCC can deal with scaling and shifting in the data axis, but not with shifting in the time axis. We address this limitation by aligning curves relative to the time of ascent of the corresponding WCB trajectories.

Figure 3a,d and Figure 3b,e show, respectively, the initial curve plots using a random ordering of variables, and the ordering with respect to the selected temporal distribution of the variable QV. Figure 3a,d shows the ordering with respect to QR. As can be seen, a number of sensitivities behave very similarly to QR and, in particular, show a significant change at the point in time where QR changes significantly. Note here that by using the absolute value of the NCC, it is ensured that parameters with high negative correlation are shown before those with low absolute correlation.

A limitation of NCC is that time series which show a similar but time-shifted behavior are found to be dissimilar (cf. Fig. 4). Even though this can be avoided by computing NCC for successively delayed versions of the original series and finding the peak in the sequence of similarities, we provide a different alternative that takes into account that it is in particular the ascent phase of a trajectory which is of interest. We define the start of the ascent of a trajectory as the start of the most rapid ascent within a 2 h window. This is calculated by using a sliding window of 2 h and calculating the total ascent within this time 185  window. Finally, the trajectories are shifted in time so that they all start their ascent at the same time, and the shifted versions are then sorted via NCC.

     To facilitate an improved comparative analysis of the sensitivities along multiple trajectories, it is furthermore important to find similar reoccurring subsequences in this data. In particular, since trajectories are seeded at different locations and times, they can first travel close to the surface over different time intervals, before similar upstream paths are observed along 190  which specific sensitivity patterns occur. To determine similar patterns, the user can select a time interval using the mouse, and automatically the subsequence of sensitivity values within this interval is searched in the same and all other curves via the subsequence matching algorithm SPRING (Sakurai et al., 2007). SPRING selects all subsequences with a dynamic time warping (DTW) distance less than a user controlled threshold, by warping one sequence so that it best matches another sequence (see Fig. 5 for a schematic illustration). The DTW distance is the sum of the per-element distances of two such optimally aligned 195  sequences. When searching for all subsequences in a sequence of length $n$ with respect to a query sequence of length $m$ with a DTW distance less than a user-specified threshold, a naive algorithm has a time complexity of $O(n^3 m)$. Due to its time complexity of $O(nm)$, SPRING enables interactive use even for long sequences.

     As SPRING is based on dynamic time warping, the time scale of subsequences may be both stretched or compressed. As can be seen in Fig. 3c,f, this enables to select, e.g., all falling edges in the temporal developments, independently of their duration. 200  The found subsequences are underlined by red background color. Compared to NSPRING (Gong et al., 2014), an extension of





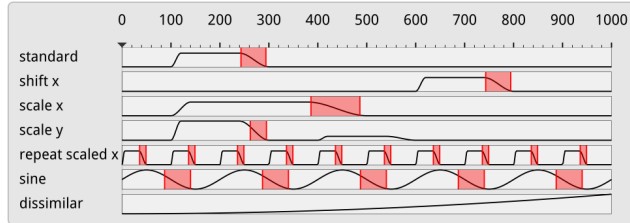

**Figure 5.** Subsequence matching in the curve plot view using SPRING. SPRING, due to dynamic time warping, can pick up patterns that are shifted and scaled in the time axis.

SPRING that adds support for data normalization, in all of our experiments SPRING gave most plausible results in line with our perception of similarity (i.e., that the similarity of two sub-sequences is also dependent on their scale).

The number of sensitivities that can be read by the system is not limited, yet beyond a certain number the corresponding curve plots cannot be shown simultaneously and the user needs to scroll through them. Especially in this case the functionality to quickly identify interesting sensitivities through similarity sorting and subsequence matching is beneficial.

## 3.2 Trajectory view

In the trajectory view, all given trajectories are shown in their geospatial context using Met.3D (cf. Fig. 2). Each trajectory is rendered as a colored and illuminated tube with black outlines to let it stand out against the background. By default, the target variable and the maximum sensitivity are encoded by two different colors, and they are shown on the tube via two bands running into the direction of the trajectory's tangent (see Fig. 6a,b for an illustration).

However, when defining these bands in object space (i.e., the assignment of points on the tube surface to either band is fixed; cf. Fig. 6a), parts of a band can disappear and become visible on the opposite surface part when rotating about the trajectory or when the tube twists. This makes it difficult to match a band with its corresponding quantity, and it is especially critical when multiple trajectories are shown and need to be compared regarding the data that is shown in the bands. To avoid this problem, we have developed a rendering technique that renders the bands so that each band covers always one half of the visible tube surface regardless of the current view and the tube's orientation (cf. Fig. 6b). This rendering is used in all trajectory views throughout this work.

While in principle it is possible to show more than two bands on each trajectory, quickly with increasing view-distance the bands cannot be distinguished anymore. To circumvent this restriction, we propose a focus view that utilizes a locally enlarged surface to obtain more space for the shown sensitivities. On each trajectory, a sphere with adjustable radius is rendered at the currently selected time. The sphere acts both as a time marker and a magnifying glass enabling the display of more sensitivities at once. By showing the focus sphere on each trajectory only at the selected time, occlusions that are introduced when increasing the radii of the trajectories everywhere can be minimized.

To show more than two sensitivities on a focus sphere, we introduce two different visual mappings. The first mapping acts like a magnifying glass when multiple sensitivities or variables were encoded via bands on the tube and continued across



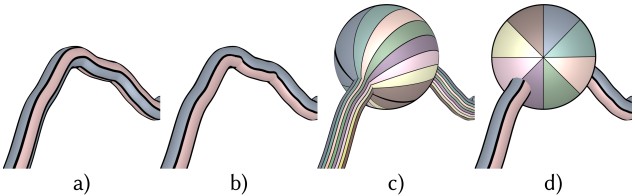

a)  b)  c)  d)

**Figure 6.** Target variable (bluish colormap) and maximum sensitivity (reddish colormap) are mapped to the trajectory surface via a) object space bands and b) view-aligned color bands. c) Multiple sensitivities are mapped to view-aligned bands running along the trajectory and across a focus sphere that enlarges the available visualization area. d) The use of consistently view-aligned pie charts improves readability of multiple sensitivities and enables an effective comparison between different trajectories.

the sphere (cf. Fig. 6c). When crossing over the sphere, the bands become wider so that the different colors can be better perceived and distinguished. As for the bands on the tube, also the bands crossing over the sphere are view-aligned, i.e., while they orient according to the trajectory tangent they cover the same parts of the visible sphere surface. The advantage of this mapping is that the alignment of the bands with the trajectories gives a visually smooth and fairly uncluttered appearance.

On the other hand, due to the illumination of the sphere surface, towards the silhouettes the bands become darker so that the relationships between colors and sensitivities are disturbed. We counteract this by reducing light-dependent shading of the bands, i.e., the coefficients of the Blinn-Phong shading model (Blinn, 1977) are reduced for diffuse and specular lighting, while being increased for ambient lighting. Nevertheless, since at some time steps spheres can become positioned at trajectory points with vastly different tangents, a visual comparison of the seen band patterns — which are then differently oriented — becomes

difficult.

The second mapping intends to avoid the aforementioned drawbacks by using a coloring that neither mimics the use of bands nor is aligned with the trajectory. Our proposed solution is the use of a pie chart-based coloring of the sphere, with each piece given equal area and colored according to a certain sensitivity (cf. Fig. 6d). The values are taken at the selected time step from the trajectory and used to color the pie pieces. The user selects the sensitivities to be shown on the pie chart, and the pie

chart is automatically subdivided into an equal number of pieces. Also the pie charts are view-aligned, i.e., they are aligned with the up-axis of the view (cf. Sect. 4). As for sphere coloring using bands, $N$ best distinguishable colors are chosen from the Brewer colormap (Harrower and Brewer, 2003). By default, we offer users the 8-class "Set1" qualitative color map from "ColorBrewer" plus turquoise. Sensitivities from low to high are mapped from 20 % saturated to fully saturated colors. This avoids that adjacent pieces with low sensitivities fade out to almost indistinguishable colors. Since each piece of a pie chart is

equally affected by shading, the use of shading is less problematic than for bands. Furthermore, each view-aligned chart has a consistent orientation, which makes it easier to compare charts on multiple trajectories.

A disadvantage is that pie charts might seem to stand apart from the trajectories, since they are not aligned with their tangents. To aid users in reading individual values off from the trajectories and spheres, a mouse hover-over is supported to inspect the values of the quantities below the mouse cursor. The use of this hover-over is demonstrated in Video 1 (04:56 min). In order

to avoid clutter and visual overload due to a high number of trajectories displayed simultaneously, we support deselecting





individual trajectories with the mouse. These trajectories are then desaturated in the 3-D view. Also, it is left up to the choice of the users to optionally use discrete, quantized color maps instead of the continuous color maps used in the figures of this work.

## 4   Implementation

All techniques presented in this paper have been integrated into Met.3D, which uses the OpenGL API for GPU-based rendering. For drawing the curve plot view, the vector graphics library NanoVG[1] is embedded. It provides hardware-accelerated rendering of vector graphics elements like anti-aliased lines and polygons, and the specification of scissor geometry to restrict rendering to a rectangular screen region. This is necessary for providing a scroll bar for the content of the curve plot view.

Met.3D offers functionality to render 3-D trajectories using illuminated polygonal tubes, including a base map showing

the earth's surface and shadows cast by the trajectories. However, the specific rendering options required by our approach, i.e., showing view-aligned bands on trajectories and spheres, as well as view-aligned pie charts on spheres, are not available. Notably, these options cannot be realized using object-space texture mapping or standard pixel shaders due to the requirement to keep the color patterns fixed in screen space.

A detailed description of our implementation is given in Appendix A and B. In the following, we outline the basic concepts

underlying the implementation, including additional rendering options.

### 4.1   View-aligned bands

For rendering the trajectories, it needs to be determined for each fragment that is rendered for the tube surface to which of the $n$ bands in screen space it belongs. Each fragment lies on a circular arc orthogonal to the trajectory tangent (cf. Fig. 7). The bands run perpendicular to this arc along the tangent direction of the trajectory. In order for the bands to have equal thickness, the

angle along the arc to the fragment position is projected onto a line perpendicular to the tangent, which removes the curvature of the arc from the individual bands. The projected arc is then subdivided into $n$ sectors which all have the same height in screen space, and the fragment is classified according to the sectors by computing its relative position $d_{band}$ in the projection and assigning the corresponding variable index $i_{var}$ to it. All required parameters can be derived solely from local properties of the rendered surface, i.e., the surface normal vector $n$, the trajectory tangent vector $t$ and the camera view vector $v$. In

particular, by projecting the camera view direction into the plane orthogonal to the trajectory's tangent direction, the problem of computing the circular arc and the angle it subtends can be reduced to a two-dimensional problem (cf. Appendix A).

### 4.2   View-aligned pie charts

To color a sphere with a pie chart that encodes the values of multiple parameters into its pieces, the screen space projection of the sphere is subdivided into a predefined number of individual pieces. To achieve a consistent assignment of parameters to

pieces for all spheres, first the angle $\alpha_{band}$ representing the angular distance of a fragment $p_{frag}$ to the up-axis of the camera

---

[1]https://github.com/memononen/nanovg





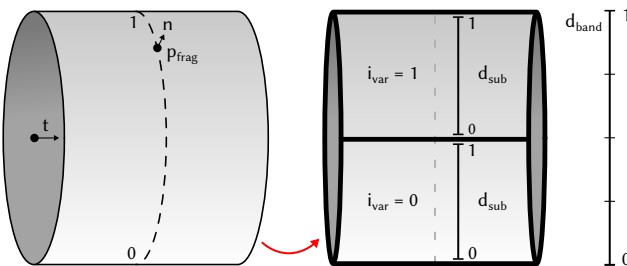

**Figure 7.** Illustration of local surface properties and subdivision of the visible part of a trajectory to determine a fragment's band position $d_{band}$, the sub-band position $d_{sub}$ and its corresponding variable ID $i_{var}$.

is computed. The global band position $d_{band}$ is then given by

$$d_{band} = \frac{\alpha_{band} \bmod 2\pi}{2\pi}. \tag{2}$$

When mapping $N$ parameters onto the sphere, the band position $d_{band} \in [0, 1)$ is subdivided into multiple sub-band positions $d_{sub}$.

## 5  Results: Case-study "Vladiana"

WCB trajectories associated with extratropical cyclone "Vladiana" ascend in a wide region near the cyclones' fronts between 23 September 2016 and 26 September 2016 (Oertel et al., 2019, 2020), where WCB ascent leads to substantial surface precipitation (Fig. 8). A 3-D view on the trajectories' ascent in the vicinity of Vladiana's fronts has recently been provided by Beckert et al. (2023). Here, we demonstrate the value of our new visual analysis method by discussing first investigations of the sensitivity of the rain mass density (QR) to microphysical parameters along WCB trajectories within Vladiana. For the example presented here, we are interested in the comparison of sensitivities related to QR along trajectories in (i) different regions of the cyclone and (ii) for WCB trajectories with different ascent behavior, i.e., we are particularly interested in the spatial variability of sensitivities and their relation to the WCB ascent rate. The interactive aspects of the analysis are documented in the Supplementary Video 2.

We focus on selected subsets of trajectories to analyze the joint development of multiple sensitivity parameters. To pre-select different group of trajectories, the 8744 available WCB trajectories have been clustered with k-means into different groups (cf. Fig. 8). We use the location and ascent rate of WCB trajectories as distinction criteria for the clustering to analyze the spatial dependencies of parameter sensitivities (Q3) and the characteristics for different types of trajectories (Q5). From the clusters, we further select five trajectories with the slowest and five with the fastest ascent in the north and south, respectively.

Figure 8 and Video 2 (00:44 min) illustrate the substantially different ascent behavior of the fast compared to the slowly ascending WCB trajectories and simultaneously show that QR is primarily important during the ascent of WCB air parcels. In the following, we analyse the sensitivity of QR to microphysical parameters and compare the multi-parameter sensitivities (i) in trajectories ascending in the north and south, and (ii) across fast and slow trajectories.





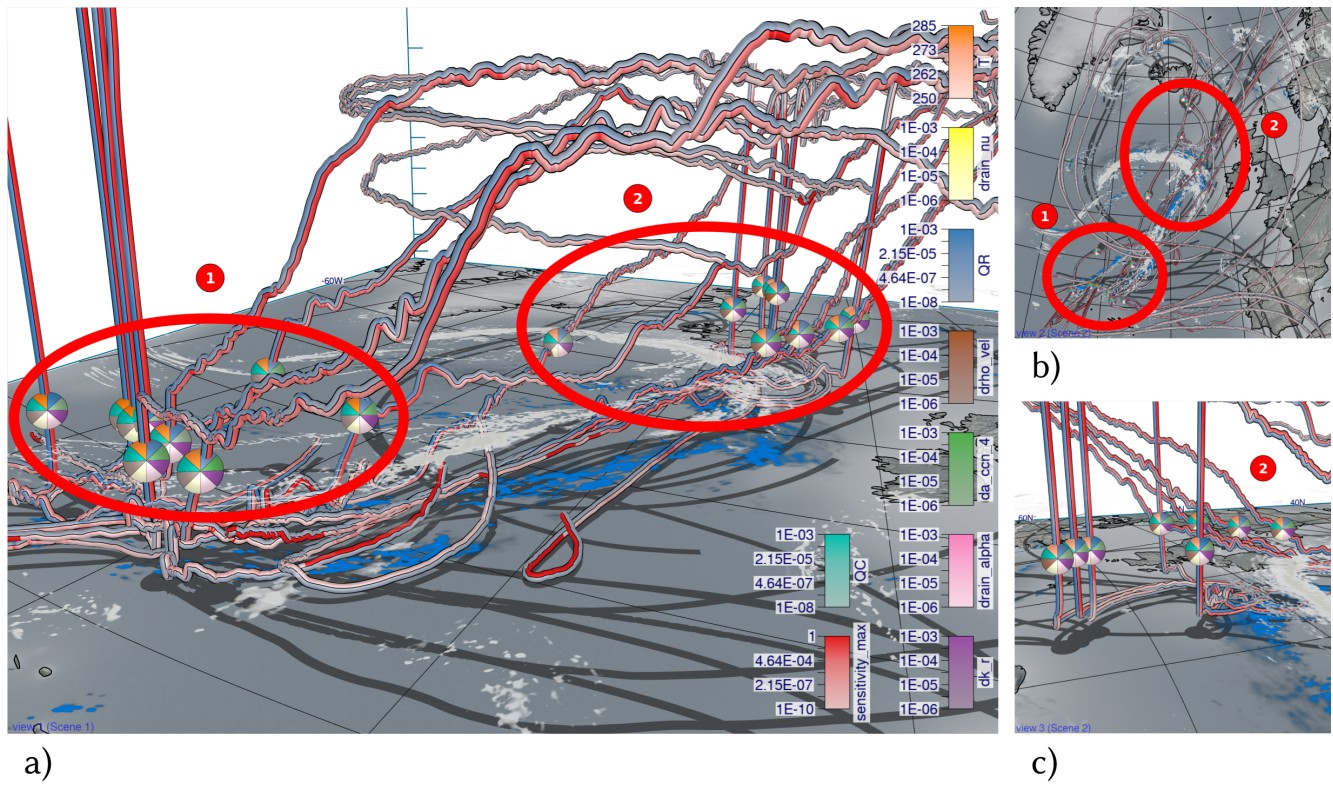

**Figure 8.** Overview of selected trajectories and first insights with spheres at the same height. Low-level clouds at approximately 1500 m altitude (gray) and surface precipitation (blue) are shown at 07 UTC 23 September 2016 when multiple trajectories start their ascent. a) Trajectories ascending in the south (group 1) and in the north (group 2) with spheres showing eight variables each. b) View from the top with the northern group 2 near clouds and precipitation and the southern group 1 with less clouds and precipitation. c) A close-up view of group 2.

## 5.1 Spatial variability of parameter sensitivities

Figure 9 shows curve plots with trajectories selected either from the southern (Figure 9a) or northern (Figure 9b) group, to analyze and compare trends of parameters across one or more groups of trajectories (Q1, Q2, and Q3). We select QR as the target variable, and center the x-axis by the time of rapid ascent of each trajectory to understand if coherent sensitivity patterns emerge once trajectories are centered relative to their time of ascent (Q4). The variance of the sensitivities (blue shades) is similarly distributed for both groups, but peaks appear at different times. The southern group shows QR maxima

at the start of the ascent, while the northern group is characterized by larger QR maxima a few hours before the ascent starts. From this, we can infer that the variance between trajectories with different locations of ascent is higher than between trajectories with a similar location. Such high QR along trajectories can arise from either (i) sedimentation of rain from above (influenced by parameters alpha, beta, and gamma in the numerical model's parameterization) or (ii) local production of rain





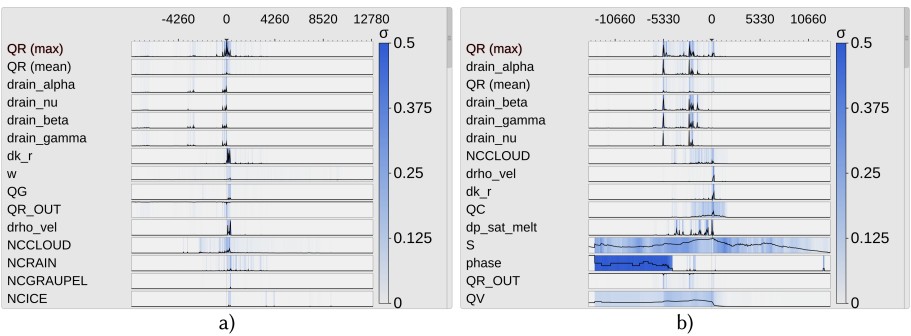

**Figure 9.** Curve plots aligned by time of ascent. The labels for the x-axis show the simulation time step, where each simulation step stands for 20 s. a) Only trajectories from the southern group have been selected. There are large peaks for rain mass density (QR) around the start of the ascent, indicating precipitation from above. b) Trajectories from the northern group with a peak in QR several hours before their ascent starts.

drops from collision of available cloud droplets (influenced by the cloud condensation nuclei (CCN), the mass density of cloud droplets (QC), and a cloud collision parameter (k_r); for a detailed description of these parameters see Seifert and Beheng (2006); Hieronymus et al. (2022)). Hence, we are interested in which processes are relevant and dominate in which region. The automated ordering (Sect. 3.1) of the parameters provides further insight (see Video 2, 03:03 min). The parameters are sorted by similarity in each time step to the maximum of QR. The sensitivities to the parameters rain_alpha, rain_beta, rain_gamma (used for sedimentation velocity), and rain_nu (used in the description of the size distribution of rain drops) are the variables with the highest similarity to QR in both cases.

Sensitivities to CCN parameters and to k_r are ranked higher in the southern group, indicating that rain drop formation due to collisions of cloud droplets is closely related to local QR formation. These correlations are not present in the northern group, which indicates that local QR maxima result from the sedimentation of precipitation from above. We conclude that QR, specifically local maxima of QR, in the southern group is more closely related to the formation of cloud droplets and subsequent conversion to rain drops than in the northern group (Q3).

To elaborate on the spatio-temporal evolution of sensitivities (Q3), we compare the maximum sensitivity of QR to any parameter in Fig. 8. The blue color along trajectories shows QR, whereas red indicates the maximum sensitivity to any parameter. Low sensitivity values (i.e., unsaturated bands) appear mostly when the trajectories descend and after they have reached their maximum height (Fig. 8a,b). This corroborates that processes influencing QR dominate during updrafts and at lower altitudes, and are generally larger for faster ascending WCB trajectories.

To further investigate which trajectories are related to large peaks in QR before the ascent starts, we use the spheres and move them slowly along the trajectories Video 2 (03:21 min). Figure 10 shows a detailed view of one such trajectory. The position of the sphere indicates the current position of the air parcel and the blue color corresponds to its QR. The blue shade on the ground is the surface precipitation shown at the same time as the air parcel location (i.e., at 81 hours simulation time). The pink color on the sphere shows the sensitivity of QR to a parameter related to the sedimentation of rain and illustrates that



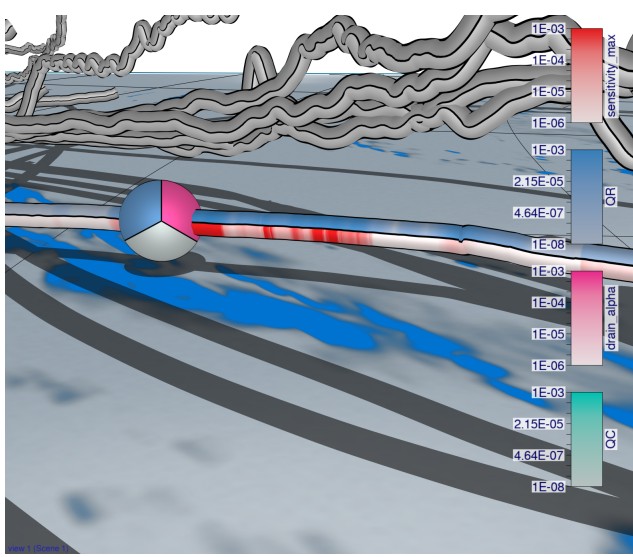

**Figure 10.** View-aligned pie chart for a selected trajectory in the lower troposphere. The sedimentation parameter drain_alpha (pink) is more pronounced where large amounts of rain mass (blue) appear and where rainfall is high (blue shade on the ground). The maximum sensitivity (red) here stems from the parameter drain_alpha. Even though the rain mass density (QR) is large, no cloud droplets are present (turquoise).

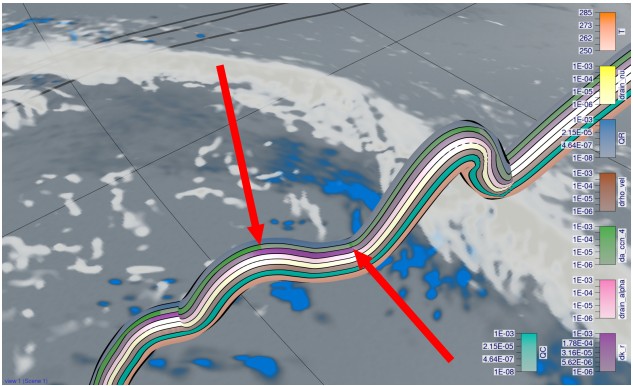

**Figure 11.** View-aligned band and pie chart with multiple sensitivities for a slantwise trajectory. Zoomed in at a slantwise trajectory. The green band (da_ccn_4 associated with cloud droplet formation) alternates with the purple band (dk_r associated with cloud droplet collision to form rain drops).

QR is particularly sensitive to the model representation of rain sedimentation in regions with high QR. Video 2 (03:21 min) shows the spatial correlation between rainfall at the surface and the peaks in rain mass density for the bands in the background of Fig. 10, which are all trajectories that started in the south and with a strong ascent in the north.





## 5.2 Influence of ascent rate on parameter sensitivities

At last, we illustrate differences in sensitivities between convective and slantwise trajectories (Q5). Generally, QR and the associated parameter sensitivities are higher along convectively ascending trajectories than along slantwise trajectories (Fig. 12; cf. Fig. 13 and Fig. 14). In the following, we illustrate examples of differences in parameter sensitivities, which are relevant for local precipitation characteristics.

First of all, QR is more sensitive to processes related to cloud droplet number concentration (da_ccn_4) and collision pro-

345 cesses (dk_r) along convective trajectories, prominently shown in Fig. 12. The color intensities of da_ccn_4 (green) along slantwise ascending trajectories (e.g., Fig. 8a) are lower than for convective ones, which indicates that processes associated with da_ccn_4 have a minor effect on QR during slantwise ascent. Similarly, the collision of cloud droplets (dk_r; purple color) is more important during convective ascent. This agrees with our previous assessment, and shows that the formation of cloud droplets and their subsequent conversion to QR are more important for QR along convective ascent than for slantwise ascent.

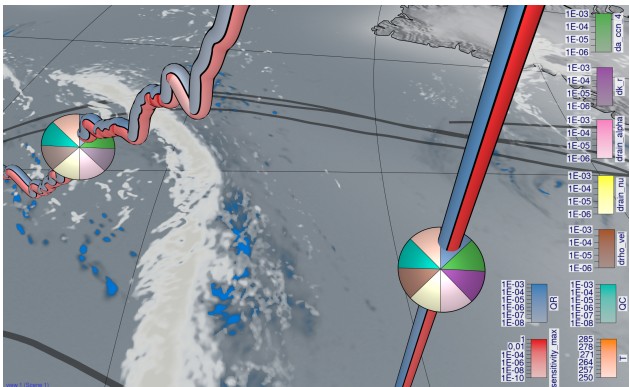

**Figure 12.** View-aligned bands with multiple sensitivities for a convective trajectory. Zoomed in at a convective trajectory with the slantwise trajectory from Fig. 11 on the left. The green (da_ccn_4 associated with cloud droplet formation) and purple (dk_r associated with cloud droplet collision to form rain drops) sensitivities have simultaneously large values during the convective ascent.

For a more detailed analysis, we zoom in to a slantwise ascending trajectory, and use multiple bands to show several parameters at once (cf. Fig. 11). Figure 11 reveals an alternating pattern between dk_r (purple) and da_ccn_4 (green). The overall slantwise ascent of the trajectory is characterized by short periods of sharp ascent with more pronounced cloud droplet formation. These periods are interrupted by periods of slower ascent and even descent, during which the collision of cloud droplets is the dominant sensitivity. These processes do not alternate in convectively ascending trajectories (Fig. 12), and instead, occur

simultaneously. This can produce and accumulate large amounts of QR quickly (cf. Fig. 8c with convective trajectories in the foreground and slantwise trajectories in the background), leading to more intense surface precipitation in a limited area. In contrast, during slantwise ascent these processes are spread over a larger area. These illustrative examples are in line with previous studies on the impact of different ascent behavior on large-scale precipitation patterns in extratropical cyclones (Oertel et al., 2019, 2020, 2021; Jeyaratnam et al., 2020).



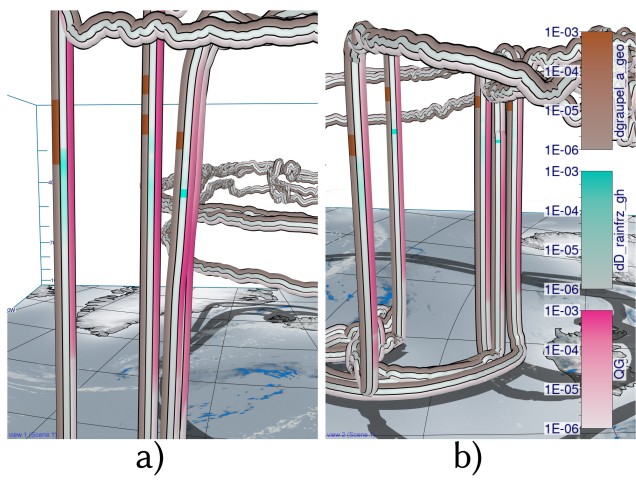

a)        b)

**Figure 13.** View-aligned bands with multiple sensitivities for convective trajectories. Convective trajectories from the southern cluster (a) and the northern cluster (b) with sensitivities to dgraupel_a_geo and dD_rainfrz_gh. Additionally, graupel mass density (QG; pink) is shown to highlight the amount of graupel that is present in convective trajectories. Sensitivities to freezing and conversion of rain to graupel and hail are visible in higher altitudes for both clusters with clear graupel formation.

As a second example, Figure 13 shows convective trajectories with sensitivities dgraupel_a_geo (determines the shape of graupel) and dD_rainfrz_gh (influences the maximum size of graupel when rain drops freeze). Large sensitivities dD_rainfrz_gh (turquoise) emerge in both the northern cluster and southern cluster. Moreover, larger sensitivities dgraupel_a_geo occur at low altitudes due to sedimentation and subsequent melting of graupel, which represents a source of QR. At higher altitudes and colder temperatures, where dD_rainfrz_gh becomes relevant (i.e., rain starts to freeze and is converted to graupel), the sensi-

tivity dgraupel_a_geo is more likely due to freezing of rain droplets. Due to the locally higher ascent velocity along convective trajectories, cloud droplets and rain drops are present at higher altitudes, which subsequently facilitates riming and graupel formation. In contrast, slantwise trajectories show hardly any sensitivity dD_rainfrz_gh or dgraupel_a_geo, if any at all, as illustrated in Fig. 14. This difference, and the difference in graupel water content between convective and slantwise trajectories, emphasizes convective trajectories' role in forming graupel and hail. These differences highlight the convective trajectories'

role for graupel formation as well as the sensitivity of QR to the model representation of riming in convective conditions.

## 6   Conclusions

We propose a novel visual analysis workflow to investigate multi-parameter properties along trajectories, here applied specifically to the relationships between the sensitivity of QR to changes in model parameters and the location and ascent behaviour of WCB trajectories. This information is required to analyze the validity of physical assumptions on which microphysical pa-

rameterizations in the source code of NWP models are based. Making the sensitivities accessible along important Lagrangian features, such as WCB trajectories, offers new insights into the correlation structures between different parameters and dif-

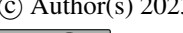


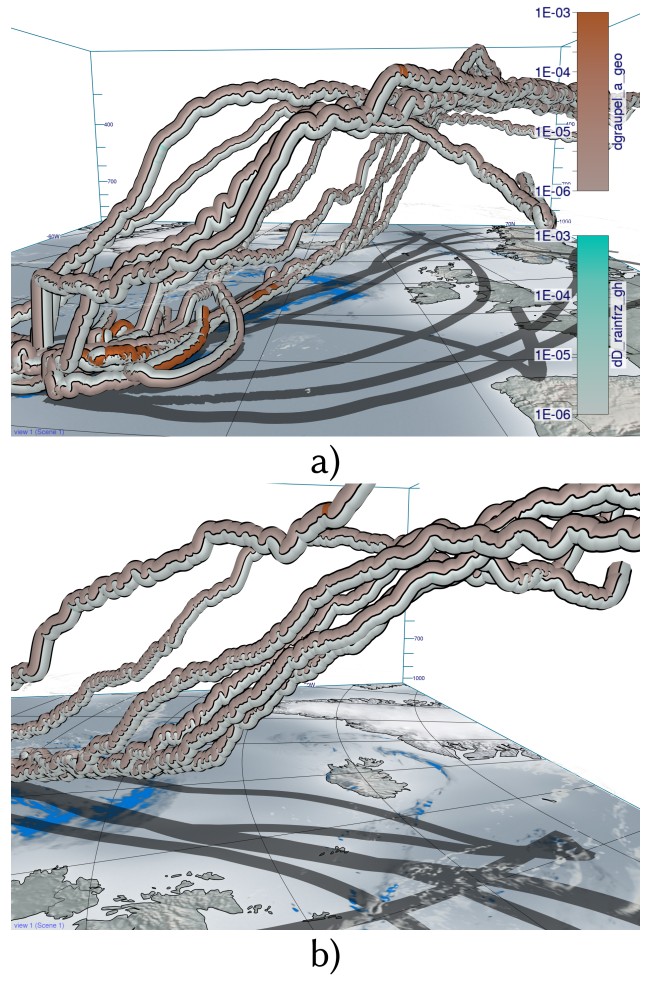

**Figure 14.** View-aligned bands with multiple sensitivities for slantwise trajectories. Slantwise trajectories from the southern cluster (a) and the northern cluster (b) with sensitivities to dgraupel_a_geo (determines shape of graupel) and dD_rainfrz_gh (determines the maximum size of graupel when rain drops freeze). Hardly any sensitivities are visible in contrast to convective trajectories in Fig. 13.

ferences between trajectories. To perform these analyses in an effective way, we link a curve plot-based summary view with a novel sphere-based focus view that enables comparison of multi-parameter distributions on different trajectories. The curve plot view provides statistical overviews and enables to quickly find parameters with similar temporal evolution. We develop the workflow in a team of scientists from visualization, high-performance computing and meteorology, and integrate it into the open-source meteorological visualization software Met.3D. The usability and benefits of the workflow are demonstrated with a real-world case-study.

Our approach can be further extended in multiple ways. First, it would be beneficial to investigate how to effectively show additional 3-D atmospheric fields, or features in these fields, in the surrounding of trajectories, to reveal specific regional multi-





field patterns causing high sensitivities. Second, the workflow could be made usable with ensembles of trajectories, where multiple sets of trajectories from different simulation runs are considered. In this way, relationships between sensitivities and the ensemble spread can be examined. Third, it would be interesting to support multiple target variables that can be switched interactively.





## Appendix A: Tube rendering

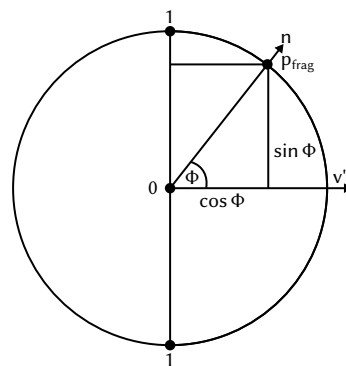

**Figure A1.** Cross section of the tube with the plane perpendicular to the tangent vector $t$ of the pathline.

To obtain a renderable trajectory representation, the trajectory (i.e., 3-D pathlines) are polygonized by extruding them into tubes in a GPU geometry shader. The parameters are mapped onto the surface of the tube as a set of bands running in the direction of the trajectory tangent (cf. Fig. 7). When mapping the bands onto the tube in object space, occlusion effects can occur, as not all parameters may lie in the front, visible part of the tube. Also, due to twist and rotations around the tube, the order in which the bands appear on screen can change and make a comparison between different tubes and the association of parameters to bands more difficult (cf. Fig. 6a). To avoid this, our rendering technique aligns the bands in view space and keeps their relative order on the screen fixed, independent of the viewing direction (cf. Fig. 6b). For this, a screen space band position $d_{band}$ is computed in the pixel shader on the GPU using only the tangent vector $t$ of the pathline associated with the tube surface fragment, the surface normal $n$ and the view vector $v = \frac{p_{cam} - p_{frag}}{\|p_{cam} - p_{frag}\|_2}$ pointing from the fragment towards the camera position $p_{cam}$ as inputs.

By projecting the camera view direction into the plane orthogonal to the tangent direction of the trajectory, the problem of computing the band position can be reduced to a two-dimensional problem. The projected camera direction $v'$ can be computed by using $v_{aux} = \frac{t \times v}{\|t \times v\|_2}$ as $v' = \frac{v_{aux} \times t}{\|v_{aux} \times t\|_2}$. The resulting setting is shown in Fig. A1.

Using the angle $\phi = \angle(v', n)$ between the projected view vector $v'$ and the normal vector $n$ would unfortunately not be sufficient as a measure, because it does not change linearly in screen space, thus producing bands of differing width. In order to derive the desired screen space measure, the fragment position needs to be projected onto an imaginary band, illustrated as the vertical line in Fig. A1. As can be seen in the figure, the normalized distance of the projected point to the center of the band amounts to the sine of the angle $\phi$. In order to compute the sine, one of the two equalities below can be used.

$$|\sin(\phi)| = \|v' \times n\|_2 = \sqrt{1 - \langle v', n \rangle^2} \tag{A1}$$

These statements hold due to the following mathematical properties of the sine, cosine, cross product and scalar product.





$$\|v'\|_2 = \|n\|_2 = 1 \tag{A2}$$

$$\langle v', n \rangle = \|v'\|_2 \|n\|_2 \cos(\phi) \tag{A3}$$

$$\|v' \times n\|_2 = \|v'\|_2 \|n\|_2 |\sin(\phi)| \tag{A4}$$

$$\sin^2(\phi) + \cos^2(\phi) = 1 \tag{A5}$$

$$\Rightarrow \|v' \times n\|_2 = |\sin(\phi)| = \sqrt{1 - \cos^2(\phi)} = \sqrt{1 - \langle v', n \rangle^2} \tag{A6}$$

As a final step, the resulting distance $|\sin(\phi)|$ needs to be corrected, as the absolute value of the sine doesn't go from 0 to 1 from one end of the imaginary band to the other, but from 1 to 0 in the middle and back to 1 at the other side. In order to correct this problem, we need to compute the sign of the sine by using the winding direction of the angle $\phi$. The sign of the sine can be computed as the sign of the volume of the parallelepiped spanned by t, v' and n.

$$\mathrm{vol}(t, v', n) = \det(t, v', n) = \langle t, v' \times n \rangle \tag{A7}$$

The equality of the determinant and the combination of the scalar product and cross product can be proven by simple expansion of the respective formulas using the three input vector coordinates as variables. Finally, we can compute the screen space band measure we are looking for as

$$d_{band} = \frac{1}{2} |\sin(\phi)| \cdot \mathrm{sgn}(\det(t, v', n)) + \frac{1}{2}. \tag{A8}$$

When mapping $N$ parameters onto the tube, we subdivide the band position $d_{band} \in (0, 1)$ into multiple sub-band positions $d_{sub}$. For this, we compute the variable ID $i_{var} = \lceil d_{band} \cdot N \rceil$ and then finally $d_{sub} = d_{band} \cdot N - i_{var}$ (cf. Fig. 7).





## Appendix B: Pie chart-based sphere rendering

For the rendering of a sphere colored via pie charts, we want to subdivide the screen projection of the sphere in angular bands, i.e., individual pie slices (cf. Fig. B1). For this, we want to compute the angle $\alpha_{band}$, which represents the angular distance of the fragment $p_{frag}$ to the up-axis of the camera. As input, we need the surface normal vector $n$, the camera view direction $v$ and the camera up-vector $up$. As a first step, the normal $n$ is projected into the view plane to obtain

$$n_{proj} = n - \langle n, v \rangle \cdot n. \tag{B1}$$

Then, we set $n' = \frac{n_{proj}}{\|n_{proj}\|_2}$. The length $\|n_{proj}\|_2$ is the normalized screen space distance to the center of the sphere. This can be easily checked for the special case $v = (0,0,1)^T$, where $\|n_{proj}\|_2$ becomes $\sqrt{n_x^2 + n_y^2} \in [0,1]$. We will use this fact later in Equation B5. In the next step, we compute the angle $\alpha_{band}$ as follows.

$$\alpha_{band} = \mathrm{atan2}(\det(n', up, v), \langle n', up \rangle) + \frac{\pi}{2} \tag{B2}$$

$\mathrm{atan2}(y, x)$ computes the angle between the positive x axis and the line connecting the origin and the point $(x, y)^T$. $\mathrm{atan2}$ returns the angle in mathematically positive direction, i.e., a counterclockwise angle. However, in our case, we do not want the counterclockwise angle to the positive x axis, but the clockwise angle from the positive y axis (the positive y axis being the up vector of the camera). This can be most easily achieved by transposing (i.e., interchanging) the x and y coordinates we feed to $\mathrm{atan2}$. To get the y coordinate of the point we use for calculating the angle, the term $\langle n', up \rangle$ is used in Equation B2. This way, we project the view plane normal onto the up axis vector. For the x coordinate, $\det(n', up, v)$ is used. We can again use Equation A7 to get the equality $\det(n', up, v) = \langle n', up \times v \rangle$. Here, $up \times v$ can be interpreted as the right axis vector of the view plane. When we project the view plane normal onto this new right axis vector, we get the x coordinate for Equation B2. The pie chart in the view plane can be seen in Fig. B1.

Finally, we can compute the global band position $d_{band}$ as

$$d_{band} = \frac{\alpha_{band} \bmod 2\pi}{2\pi}. \tag{B3}$$

When mapping $N$ parameters onto the sphere, we again subdivide the band position $d_{band} \in [0,1)$ into multiple sub-band positions $d_{sub}$ (cf. Fig. B1).

A black separator line is drawn between two neighboring sub-bands. A problem that also arises for the pie chart-based spheres is that changes in the sub-band position are not linear in screen space and dependent on the distance to the screen space center of the sphere. Consequently, two correction factors are introduced below, and the final separator thickness is computed as

$$w'_{sep} = \frac{w_{sep}}{f_1 f_2}. \tag{B4}$$

The factor $f_1$ is equal to $\|n_{proj}\|_2$, which itself, as was shown earlier in this section, is equal to the normalized distance to the screen space center of the sphere. This way, it is guaranteed that the separator thickness doesn't get thinner the closer we





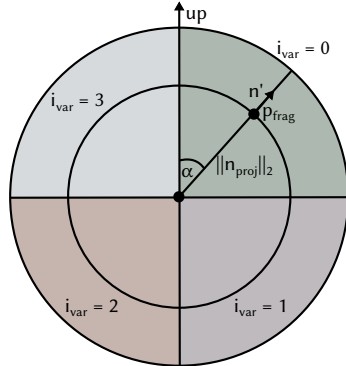

**Figure B1.** Illustration of how the input vectors and points on the sphere are used to compute the band position $d_{band}$, the sub-band position $d_{sub}$ and its corresponding variable ID $i_{var}$.

get to the center of the pie chart.

$$f_1 = \|n_{proj}\|_2 \tag{B5}$$

Finally, the factor $f_2$ is used to make sure that the separator thickness of the pie chart sphere and the trajectory tube match.
For this, the circumference of the sphere $2r\pi$ is divided by the width of the tube $w_{tube}$.

$$f_2 = \frac{2r\pi}{w_{tube}} \tag{B6}$$





**Appendix C: Variable and parameter names**



**Table C1.** Variable names in the data set.

| Variable | Description |
|---|---|
| pressure | Pressure in hPa |
| T | Temperature in Kelvin |
| w | Vertical velocity in $\mathrm{m\,s^{-1}}$ |
| S | Saturation |
| QV | Water vapor mass density in $\mathrm{kg\,m^{-3}}$ |
| QC | Cloud mass density in $\mathrm{kg\,m^{-3}}$ |
| QR | Rain mass density in $\mathrm{kg\,m^{-3}}$ |
| QS | Snow mass density in $\mathrm{kg\,m^{-3}}$ |
| QI | Ice mass density in $\mathrm{kg\,m^{-3}}$ |
| QG | Graupel mass density in $\mathrm{kg\,m^{-3}}$ |
| QH | Hail mass density in $\mathrm{kg\,m^{-3}}$ |
| NCCLOUD | Cloud number density in $\mathrm{m^{-3}}$ |
| NCRAIN | Rain number density in $\mathrm{m^{-3}}$ |
| NCSNOW | Snow number density in $\mathrm{m^{-3}}$ |
| NCICE | Ice number density in $\mathrm{m^{-3}}$ |
| NCGRAUPEL | Graupel number density in $\mathrm{m^{-3}}$ |
| NCHAIL | Hail number density in $\mathrm{m^{-3}}$ |
| QR_OUT | Sedimentation of rain mass density out of the air parcel in $\mathrm{kg\,m^{-3}}$ |
| QS_OUT | Sedimentation of snow mass density out of the air parcel in $\mathrm{kg\,m^{-3}}$ |
| QI_OUT | Sedimentation of ice mass density out of the air parcel in $\mathrm{kg\,m^{-3}}$ |
| QG_OUT | Sedimentation of graupel mass density out of the air parcel in $\mathrm{kg\,m^{-3}}$ |
| QH_OUT | Sedimentation of hail mass density out of the air parcel in $\mathrm{kg\,m^{-3}}$ |
| NR_OUT | Sedimentation of rain number density out of the air parcel in $\mathrm{m^{-3}}$ |
| NS_OUT | Sedimentation of snow number density out of the air parcel in $\mathrm{m^{-3}}$ |
| NI_OUT | Sedimentation of ice number density out of the air parcel in $\mathrm{m^{-3}}$ |
| NG_OUT | Sedimentation of graupel number density out of the air parcel in $\mathrm{m^{-3}}$ |
| NH_OUT | Sedimentation of hail number density out of the air parcel in $\mathrm{m^{-3}}$ |
| latent_heat | Latent heat released by cloud microphysical processes in $\mathrm{J\,kg^{-1}}$ |
| latent_cool | Latent heat absorbed by cloud microphysical processes in $\mathrm{J\,kg^{-1}}$ |
| z | Height in m |
| Inactive | Number of nuclei that can not be activated for ice, snow, graupel or hail |
| deposition | Mass density of water vapor deposited in ice, snow, graupel and hail |
| sublimination | Mass density of water vapor from ice, snow, graupel and hail |
| time_after_ascent | Time centered to the start of the fastest ascent in a 2 h time window |





| Variable | Description |
| --- | --- |
| conv_400 | Flag for a convective ascent of 400 hPa |
| conv_600 | Flag for a convective ascent of 600 hPa |
| slan_400 | Flag for a slantwise ascent of 400 hPa |
| slan_600 | Flag for a slantwise ascent of 600 hPa |
| step | Simulation step |
| phase | Flag for different phases of the trajectory. 0: warm phase, 1: mixed phase, 2: ice phase, 3: neutral phase |





**Table C2.** Parameter names in the data set.

| Parameter | Description |
| --- | --- |
| inv_z | Inverse of air parcel size (height) used in explicit sedimentation) (cf. Hieronymus et al. (2022)) |
| rho_vel | Exponent for density correction in velocity-mass-relations (cf. Seifert and Beheng (2006), Eq. (33)) |
| D_rainfrz_gh | Size threshold for partitioning of freezing rain in the hail scheme (cf. Seifert and Beheng (2006)) |
| p_sat_melt | Saturation pressure at $T = 273.15\,\mathrm{K}$ (cf. Seifert and Beheng (2006)) |
| a_HET | Exponent for heterogeneous rain freezing with data of Barklie and Gokhale (cf. Seifert and Beheng (2006)) |
| k_r | Coefficient for accretion of QC to QR (cf. Seifert and Beheng (2006)) |
| a_ccn_1 | Parameter for CCN concentration (cf. Hande et al. (2016)) |
| a_ccn_4 | Parameter for CCN concentration (cf. Hande et al. (2016)) |
| b_ccn_1 | Parameter for CCN concentration (cf. Hande et al. (2016)) |
| b_ccn_3 | Parameter for CCN concentration (cf. Hande et al. (2016)) |
| b_ccn_4 | Parameter for CCN concentration (cf. Hande et al. (2016)) |
| c_ccn_1 | Parameter for CCN concentration (cf. Hande et al. (2016)) |
| c_ccn_3 | Parameter for CCN concentration (cf. Hande et al. (2016)) |
| c_ccn_4 | Parameter for CCN concentration (cf. Hande et al. (2016)) |
| d_ccn_1 | Parameter for CCN concentration (cf. Hande et al. (2016)) |
| d_ccn_2 | Parameter for CCN concentration (cf. Hande et al. (2016)) |
| d_ccn_3 | Parameter for CCN concentration (cf. Hande et al. (2016)) |
| d_ccn_4 | Parameter for CCN concentration (cf. Hande et al. (2016)) |
| rain_a_geo | Coefficient for diameter size calculation (cf. Seifert and Beheng (2006) Eq. (32)) |
| rain_b_geo | Exponent for diameter size calculation (cf. Seifert and Beheng (2006) Eq. (32)) |
| rain_min_x | Minimum size of the particle used after the microphysics (cf. Seifert and Beheng (2006), Eqs. (94), (97)) |
| rain_a_vel | Coefficient for particle velocity (cf. Seifert and Beheng (2006) Eq. (33)) |
| rain_b_vel | Exponent for particle velocity (cf. Seifert and Beheng (2006) Eq. (33)) |
| rain_alpha | Constant in rain sedimentation (cf. Seifert (2008), Eq. (A10)) |
| rain_beta | Coefficient for rain sedimentation (cf. Seifert (2008), Eq. (A10)) |
| rain_gamma | Exponent for rain sedimentation (cf. Seifert (2008), Eq. (A10)) |
| rain_nu | Parameter to calculate the shape of the generalized $\Gamma$-distribution (cf. Seifert and Beheng (2006), Eq. (79)) |
| rain_mu | Shape parameter of the generalized $\Gamma$-distribution (cf. Seifert and Beheng (2006), Eq. (79)) |
| graupel_a_geo | Coefficient for diameter size calculation (cf. Seifert and Beheng (2006) Eq. (32)) |
| graupel_b_geo | Exponent for diameter size calculation (cf. Seifert and Beheng (2006) Eq. (32)) |
| graupel_a_vel | Coefficient for particle velocity (cf. Seifert and Beheng (2006) Eq. (33)) |
| graupel_b_vel | Exponent for particle velocity (cf. Seifert and Beheng (2006) Eq. (33)) |
| graupel_vsedi_max | Maximum sedimentation velocity parameter (cf. Hieronymus et al. (2022)) |
| ice_a_geo | Coefficient for diameter size calculation (cf. Seifert and Beheng (2006) Eq. (32)) |



| Parameter | Description |
|---|---|
| ice_b_geo | Exponent for diameter size calculation (cf. Seifert and Beheng (2006) Eq. (32)) |
| ice_b_vel | Exponent for particle velocity (cf. Seifert and Beheng (2006) Eq. (33)) |
| ice_vsedi_max | Maximum sedimentation velocity parameter (cf. Hieronymus et al. (2022)) |
| snow_b_geo | Exponent for diameter size calculation (cf. Seifert and Beheng (2006) Eq. (32)) |
| snow_b_vel | Exponent for particle velocity (cf. Seifert and Beheng (2006) Eq. (33)) |
| snow_vsedi_max | Maximum sedimentation velocity parameter (cf. Hieronymus et al. (2022)) |

*Code and data availability.* The implementation of the visualization techniques described in this work is available in a fork of the open-
470 source 3-D visualization system Met.3D at https://github.com/chrismile/met.3d under the terms of the GNU General Public License v3.0
(GPL-3.0). Version 1.6.0-multivar0 of this software is archived at (Neuhauser et al., 2023). The trajectory data used for the realization of
the figures and the case study is archived at (Hieronymus and Oertel, 2023) under the terms of the Creative Commons Attribution 4.0
International License. The algorithmic differentiation code used for the generation of this data is described at (Hieronymus et al., 2022) and
made available at (Hieronymus, 2022) under the terms of the MIT License.

*Video supplement.* Two video supplements showcasing the functionality of the visualization techniques described in this work (Video 1) and
illustrating the analysis of the "Vladiana" WCB trajectories (Video 2) are available at (Neuhauser and Hieronymus, 2023).

*Author contributions.* CN, MH, MR, AO and RW wrote the manuscript. CN implemented the proposed visualization methods, wrote the
sections of the manuscript regarding the methodology and created the figures except for those for the case study. RW, CN and MK conceived
the idea. MH provided the simulated trajectory data. MH, MR and AO performed the case study and the meteorological analyses. RW
supervised the work.

*Competing interests.* The authors declare that they have no conflict of interest.

*Disclaimer.* Publisher's note: Copernicus Publications remains neutral with regard to jurisdictional claims in published maps and institutional
affiliations.

*Acknowledgements.* The authors acknowledge support by the Deutsche Forschungsgemeinschaft (DFG) within the Transregional Collabo-
485 rative Research Centre TRR165 Waves to Weather, (wavestoweather.de), Projects A7, Z2, B8 and C9 as well as funding from JGU Mainz.



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
