# Peer review of "Visual analysis of model parameter sensitivities along warm conveyor belt trajectories using Met.3D (1.6.0-multivar1)"

_Geoscientific Model Development, 2023_

## Author Comment (AC1)

**1   Author's Response**

Dear referee,

We would like to thank you for the in-depth review of our work. After the end of the discussion period, we will post a revised version of our manuscript which takes into account the recommended suggestions for improvement. In the following sections, we will respond to the individual points raised in the review. The comments from the review will be highlighted in bold, followed by our responses.

Best regards,
The authors

[Figure]

Figure 1: Updated version of Fig. 1 of the manuscript with the new visual mapping for the spheres.

[Figure]

Figure 2: Target variable (bluish colormap) and maximum sensitivity (reddish colormap) are mapped to the trajectory surface via a) object space bands and b) view-aligned color bands. c) When multiple variables are mapped to view-aligned bands running across an enlarged focus sphere, the band's distortions and alignment with the trajectory's tangent prohibit an effective visual analysis and comparison between different trajectories. d) The use of consistently view-aligned polar color charts improves readability of multiple variables and enables an effective comparison between different trajectories. While in d) values are encoded by saturation, in e) a polar chart using the radius instead of the saturation for encoding the individual values is used.

**1.1 Pie charts**

**Instead of encoding a quantitative attribute by the saturation of a fixed-size pie segment, I would suggest to consider using a polar area chart instead. The polar area chart likewise uses pie segments with fixed angle, but the radius of the segment is varied according to the quantitative attribute. Thereby, the quantitiative data is mapped to a positional encoding (i.e., the radius), which ranks higher on the effectiveness scale, and where differences between different radii can be perceived easier. Since small values lead to tiny segments, it would be imaginable to use a full glyph with maximal radius, as it is done now, which is depicted with the desaturated hue. And then, each pie segment could be filled from the center towards the outside by a fully saturated segment where the radius of the fully-saturated segment encodes the quantitative data. The saturation of a pixel could be calculated in the fragment shader based on the distance to the glyph center and the quantitative value that is to be encoded. If I'm not mistaken, it should be straightforward to implement this shading on the existing spherical glyphs. I would leave it to the authors to decide if they want to try this alternative encoding or whether they would like to discuss the perceptual limitations**

**of their current saturation-based encoding.**

- We thank the reviewer for this valuable hint. In fact, we agree that the polar area chart can be more effective for a comparative analysis as intended in our work. The system now provides this alternative, and we compare it to the saturation-based encoding (cf. Fig. 2). We have decided to not color the background of the polar area chart by the low saturation colors, as this can generate a cluttered and distracting visualization. The updated version of Fig. 1 of the manuscript can be seen in Fig. 1.

**The glyph was called a "pie chart", which suggests that a quantitative attribute is mapped to the slice angles, which was not the case. I would suggest using a different name for the glyph, in order to avoid the usually negative associations related to the inaccurate angle perception of pie charts.**

- We have considered this suggestion and renamed this type of visual mapping to *polar chart*.

**The meaning of the slice hue was not clear from the beginning. Please add color bars as in Fig. 8 also to Figs. 1 and 2.**

- We have added color bars to Figs. 1 and 2 of the manuscript.

**1.2 Preattentive encoding on 3D trajectories**

[Figure]

Figure 3: Data from Fig. 1 with predicate-based filtering where regions are highlighted where $QR \leq 10^{-3}$ and $dinv\_z \geq 0.1$.

**Along the 3D trajectories, multiple quantitative attributes are color-mapped, each with its own single-hue color map. While the individual encodings are pre-attentive, the task of finding locations in the image, where both encodings have high values at the same time, or where one is high and the other is low, is not pre-attentive anymore. If the user wants to find places, where two variables have high values at the same time, then it would be better to highlight those areas on the trajectory with a separate pre-attentive encoding, for example by high-lighting such regions. This could be considered if this task should be further supported. Since the displayed set of trajectories is usually relatively small, it is not a crucial improvement.**

- We consider this suggestion useful for accelerating the search task, especially if larger numbers of trajectories are shown. Thus, the system now provides the possibility to select value ranges of interest for two variables or sensitivites, and to fade out all locations where the values are not within these ranges. In addition, the user can select showing only those locations where both variables/sensitivites are within the selected ranges. "Fading-out" is realized by completely desaturating the segments of the trajectory with values outside of the selected ranges in order to draw the attention of the user to the regions of interest. While we do not need this feature for our case-study with a limited amount of trajectories, we demonstrate the functionality in a new subsection (cf. Fig. 3).

**The distortion of the oriented patterns on the sphere impedes the effectiveness (Fig. 6c). This is perhaps not a crucial feature and could possibly be removed.**

- Please note that the visual mapping in Fig. 6c is only shown to demonstrate that it impedes the effectiveness, and to motivate the mapping in Fig. 6d. We have revised the text and caption to make this clear. The new version of the figure can be seen in Fig. 2.

**1.3 Curve plots**

**Please add little labels for the x-axis to indicate that this axis encodes time.**

- We have added a label for the x-axis in the curve plot view.

**The curves were simplified using LTTB if there are more vertices than pixels. It was mentioned that this reduces the performance penalty of drawing too many points. I would be curious by how much the performance is improved by this optimization.**

- We now provide information on the performance improvement when using LTTB. We have noticed that, for our approach, the expected rendering time scales almost linearly with the number of drawn points. We attribute this to the fact that the vector graphics library we use, i.e., NanoVG, reassembles the drawn geometry whenever there is an interaction with the curve plot. While GPU rendering is generally quite fast and should not struggle with the amount of geometry rendered in the curve plot view, the assembly step on the CPU is proportional to the number of drawn points, which can be significantly reduced using LTTB.

**Instead of mean (using a curve) and the stdev (using color-coding), one could display the full distribution of all trajectories in a trajectory density plot.**

- We agree that density plots could help to better show the position of outliers in the data. This, however, introduces the following problems. Firstly, the vertical space of the individual curve plots is quite small, also limiting the effectiveness of a kernel plot. Secondly, when showing the distribution as a density plot, a problem is that during sensitivity analysis we usually only show a limited number of trajectories simultaneously. Consequentially, the density plot could become quite sparse. Due to this drawback, we have decided to rather stay with the statistical summary view based on the mean and stdev for the time being.

**It was mentioned that the scalability of the approach is limited when there are too many model parameters to display. It would be imaginable to achieve scalability by including table lenses. A table lense distinguishes between selected and non-selected model parameters. The non-selected model parameters would encode less information in a box with smaller height, while the selected ones encode more information in a box with larger height. Implementing this is not a necessity, but it could be mentioned that standard approaches exist to achieve scalability of the presented method.**

- A reference to table lenses is added. While we see table lenses as an effective means to show all data rows simultaneously, we are not convinced it would be beneficial in our scenario: Firstly, rows representing non-selected variables become quite thin, which is problematic in our case where we show curves requiring sufficient space on the y-axis. Secondly, similarity sorting orders all curves regarding their similarity to a selected one. By using table lenses, a certain "dissimilarity" threshold would need to be selected, beyond which all curves would be considered equally dissimilar and drawn as curves with lower height. We believe that showing all curves at the same level of detail and enabling scrolling down to the less similar curves is more effective.

**1.4  3D View**

**In Figure 1, I find it difficult to estimate the altitude of the surface precipitation. From the text, I would think that it should be on the ground, but since the drop shadows are not cast onto the surface precipitation it appears to float above the ground. Perhaps the drop shadows could also be cast onto the color-coding on the surface.**

- We agree that it could be useful in some cases to have drop shadows from one horizontal cross section onto another. We have discussed this problem with our domain experts from meteorology, who raised the concern that drop shadows on the surface precipitation could potentially falsify the values read from the visualization. Consequentially, while we are aware of the problem described by the reviewer, we have decided not to add any shadows cast onto the color-coding of the surface precipitation. Instead, in the revised manuscript we have now added shadows cast by the cross sections onto the ground, which should also alleviate the problem described by the reviewer.

**Is the vertical scaling of the clouds and the WCB trajectories equal? The clouds seem rather flat compared to the high vertical ascent of the WCBs.**

- We are unsure whether the clouds being flat refers to the flatness of the layer or the low height of the layer. We visualize the surface precipitation on the ground (kg m$^{-2}$) at a pressure level of 1045 hPa in blue and the specific cloud liquid water content in the air (kg kg$^{-1}$) at 860 hPa in white using horizontal cross sections. The same mapping from pressure to height as for the trajectories is used. The trajectories range from 1020 hPa to 219 hPa. Taking the logarithmic mapping from pressure to height into consideration, the clouds cross section is not too far from the ground. However, the cloud layer may have appeared a bit closer to the ground in the old manuscript than it actually is. In the new manuscript, drop shadows are now cast onto the earth surface. We would like to note that support for volumetric cloud rendering instead of flat horizontal cross sections is also available in Met.3D.

**1.5  Meaning of derivative activations**

**When we see on the trajectory that at a certain location the sensitivity to a certain variable is high, does this immediately mean anything, or could it be that this particular variable is actually not present? For example, for a variable to actually exhibit a change, many other conditions might have to be right. Couldn't it be that the change that would have a strong influence on QR would actually never occur in a certain weather situation? Or are certain changes on one variable easier to occur than certain changes on another variable? If this is the case, is there some room for an extension of the diagnostic tool in the future?**

- We simulate the cloud microphysics and evaluate the sensitivities with the exact ambient conditions the trajectories provide. If we changed a parameter during the simulation at the time step at which the sensitivity of QR to this parameter is large, then QR would change accordingly. We see no sensitivity when the corresponding process does not appear in the given environment or a low one if the impact is small. At this moment, it is impossible to evaluate sensitivities regarding the complete dynamical system, i.e., feedback between the dynamical core and cloud microphysics is not considered.

**1.6 Literature on trajectory data visualization**

We have added references to the suggested literature.

**1.7 Formal description**

**The input to the method could be described a bit more formally [...]**

- We have revised the manuscript accordingly.

**Also consider using a single terminology, i.e., decide on ″bands″ or ″sectors″, instead of mixing the terms.**

- In the manuscript, we use the term *bands* to refer to the colored stripes running in the direction of the tangent along the trajectory tubes and the (rejected) visual mapping shown in Fig. 6c. With sectors, we refer to the polar chart sectors. While conceptually very similar, we believe that also referring to the bands as sectors (or vice versa) might not well describe the difference of the two concepts. If the reviewers believe that using one unified term is preferable, we will be happy to realize this change in a revised version of the manuscript. When referring to the polar charts, we now consistently refer to *sectors* instead of *bands*.

**The symbol ″n″ is ambigiously used, namely for the number of bands and for the normal (also in the appendix). I suggest adding a vector (\vec) over the normal and the tangent. [...] In appendix B, I would suggest to call the ″up″ vector ″\vec k″ or something else with only one letter.**

- We agree that clarification is necessary here. Unfortunately, we were not able to implement these suggestions completely. The LaTeX command ″\vec″ has a slightly different effect in the GMD template. According to the guidelines, "vectors [are typeset] in bold-face italics", which seems to be the effect of the command in the template. We hope that typesetting all vectors in this way will still disambiguate symbols used as a scalar number and a vector-valued variable on different occasions. We also made sure to consistently use an upper-case "N" for the number of bands or sectors.

[Figure]

Figure 4: Fig. 12 from the manuscript (left old, right new).

**1.8 Case study**

**It was said that "location and ascent rate" of WCB trajectories were used as distinction criteria for the clustering. Have those two scalars been added in a weighted combination of some sort? If so, what were the weights?**

- These two criteria (location and ascent rate) are not used in a weighted way. First, we pre-select two clusters, one in the south and one in the north. Then, we select the five trajectories with the slowest and the five with the fastest ascent in the north and south, respectively. We have added a sentence to the paper describing that no weights are used.

**The bending of lines in Figure 11 looks in some cases a bit odd. If there is a problem with this encoding along lines with high curvature, it might be worth mentioning this as limitation.**

- In the previous version of the manuscript, the geometric subdivision of the trajectory for Fig. 11 was quite low. We have improved this in the new manuscript and the bending at high curvature now looks less weird (cf. Fig. 4). Still, a discontinuity can be seen at the lower left of the figure that cannot be attributed to the geometric subdivision, but self-intersections at higher curvature. This can happen when the radius for the trajectory tube is chosen too high for the local curvature, as is the case for the close-up view in Fig. 11. This is not necessarily a downside of our approach, as it also affects any other trajectory rendering technique. If no attributes are mapped onto the surface of the tube, however, this only reflects as a discontinuity in the lighting of the surface.

**Several figures could be made a bit larger (from Fig 10 onwards).**

- We agree that, especially for Fig. 11 and 12, the ratio of figure size and font size is too small, which is improved in the new submission by using a larger font size. The GMD LaTeX template for review uses a single column. If we are informed correctly, PDFs generated of the paper after the review process will use a double column format. Figures that currently take up only half of the page are supposed to then fit into a single column of the two column layout used after the review process.

**In Figure 12, I find it difficult to see a difference in da_cnn_4 for the slantwise and convective trajectories. The green shades look rather similar to me.**

- That is correct, the green shade is indeed very similar. This is intentional, as da_cnn_4 is high if dk_r is low and vice versa for the slantwise ascent as shown in Fig. 11. But only for the convective ascent both da_cnn_4 and dk_r have high values simultaneously. The description in Fig. 12 has been revised to emphasize the difference.

---

## Author Comment (AC2)

**1 Author's Response**

Dear referee,

We would like to thank you for the in-depth review of our work. In the following sections, we will respond to the individual points raised in the review. The comments from the review will be highlighted in bold, followed by our responses. After the end of the discussion period, we will post a revised version of our manuscript which takes into account the recommended suggestions for improvement.

Best regards,
The authors

[Figure]

Figure 1: Updated version of Fig. 1 in the manuscript with the new visual mapping.

[Figure]

Figure 2: Target variable (bluish colormap) and maximum sensitivity (reddish colormap) are mapped to the trajectory surface via a) object space bands and b) view-aligned color bands. c) When multiple variables are mapped to view-aligned bands running across an enlarged focus sphere, the band's distortions and alignment with the trajectory's tangent prohibit an effective visual analysis and comparison between different trajectories. d) The use of consistently view-aligned polar color charts improves readability of multiple variables and enables an effective comparison between different trajectories. While in d) values are encoded by saturation, in e) a polar chart using the radius instead of the saturation for encoding the individual values is used.

**1.1  Pie charts**

**I find the term "pie charts" quite confusing, as I first expected the slices to have different sizes to illustrate relative sizes of data. Furthermore, I find it rather difficult to capture the values in the pie charts based on the colour saturation, could you make the difference between low and high values larger?**

- The criticism regarding the term *pie charts* has been raised by both reviewers. We agree with the reviewers that the term pie chart is closely entangled with using the pie angle for the visual mapping. Consequentially, we have considered this suggestion and renamed the type of visual mapping to *polar chart*.

- As suggested by reviewer 1, we have added an improved visual mapping to our system that we call *polar area chart*, which we compare to the saturation-based encoding (cf. Fig. 2). In this new encoding, differences in value are more strongly perceivable, as quantities are mapped to the radius.

**It is difficult to understand Fig. 1 at this point in the paper, without watching the supplementary videos and before reading sections 2 and 3. Please add further details in the**

[Figure]

Figure 3: Updated version of Fig. 10 of the manuscript with an arrow pointing out the lack of cloud droplets (QC).

**caption: What is the starting time of the WCB trajectories? What is QR (it has not yet been defined when Fig. 1 is mentioned) and what are the units of QR? What is sensitivity_max and what are the units of the sensitivities? What is shown in the curve plots, i.e., what is shown on the x and y axes, what are the black lines and the blue shading? What do the colours in the pie charts show? Please add colour bars to the figure. Even with the added information, you might consider moving the plot to the data and method section (where you nicely explain the different fields) to make it easier understandable, or to mention that the details will be explained in the data and method section.**

- The updated version of Fig. 1 of the manuscript with color bars and the new visual mapping can be seen in Fig. 1. We have added color maps as requested by the reviewer. We will revise the figure description in the new manuscript accordingly, to make clear the start time of the trajectories, the units of QR, the units of the sensitivities, and what is shown in the curve plots and pie charts. We decided not to move the figure to the data and methods section, as we hope it can act as a teaser figure showing the reader the major contributions of the manuscript.

**Abstract and conclusions: Please also include some key findings about your case study. In the conclusion it would be nice if you could come back to the questions posed in the introduction and summarise the results.**

- We added some key findings in the abstract and a paragraph in the conclusion that refer to the different questions raised.

**Data, first paragraph: Over which time interval are the WCB trajectories calculated (and shown in the figures)?**

- The trajectories are calculated from 20 September 2016 at midnight until 24 Sep 2016 at 16:00 hours. Different trajectories are started every two hours until 22 Sep 2016 at 16:00 hours. The starting time for the trajectories from Section 5 covers the same range. The other trajectories showcasing different visual analysis methods start on 20 September 2016 at 00:30 hours and are calculated until 22 September 2016 at 08:00 hours. We added the information in the first paragraph of the data section.

**Line 321: "Sensitivities to CCN parameters and to k_r are ranked higher in the southern group." I see the symbol dk_r in Fig. 9a, but where are the sensitivities to the CNN parameters? I don't see any of the CCN parameter names from Table C2 in Fig. 9a.**

- Thank you for pointing this out. We inadvertently set the cut-off for the figures too high. This has been fixed by showing more rows.

**Fig. 9: From the text in the figure caption, one might infer that precipitation from above is more important for the southern than for the northern group, but according to the text it is the other way around.**

- Indeed, there has been a mistake in the caption. The large peaks of rain mass density stem from the formation of rain droplets which is also indicated by the peak of the collision parameter dk_r. We corrected the caption accordingly.

**Fig. 10: I find it confusing that you refer to the absence of something in turquoise in the caption. Should the turquoise cloud droplets (or their absence) be found in the pie or in the underlying map? If it should be in the pie, maybe you could highlight the outer border of this (grey) segment with turquoise lines. Otherwise, the reader does not know to which variable the grey part of the pie belongs.**

- We agree that the absence of color is confusing. The cloud droplets refer to the grey segment in the pie. We added an arrow and some text in Fig. 3 to highlight the lack of cloud droplets. However, highlighting the outer border of this segment works only for a single or a few pie charts. If we zoom out and compare multiple trajectories, e.g., in Fig. 1, it is easier to distinguish trajectories with and without certain sensitivities without colored borders.

**You mention Fig. 12 before Fig. 11, could you exchange them?**

- Thank you for pointing that out. We exchanged both figures (now Fig. 12 and Fig. 13).

**Line 355: "... convective trajectories in the foreground and slantwise trajectories in the background" – Do you mean group 1 and 2, respectively?**

- These are all trajectories from group 2. Both groups contain slantwise and convective trajectories, but their fastest ascent is located at similar places within each group. We added the group affiliation to the text.

**Figs. 10, 11, 12, 13, 14 (and in other places in the paper): It would make the text easier understandable if you could be more specific with the wording and more often write "sensitivities of QR to xxx" (in the figure caption and in the text).**

- At the beginning of Section 5, we clarified that the prefix "d" in the names of parameters refers to the sensitivity of QR to the parameter. We made the caption in those figures clearer and added the phrase "sensitivity of QR to" throughout the text.

**Typos and wording**
- Appendix Table C2, description of inv_z: parenthesis sign ")" after "sedimentation" not needed.
- Line 175: "Figure 3a,d shows ...": Do you mean Figure 3c,f?
- Line 296: "different group": different groups
- Line 326: "..., we compare the maximum sensitivity of QR to any parameter in Fig. 8." I find this phrase difficult to understand. Maybe better something like "..., we investigate where along the trajectories any of the parameters is associated with the maximum sensitivity (Fig. 8)."

**- Line 332: Parenthesis should include "Video 2".**
**- Line 345: "more sensitive ...  along convective trajectories ..." – Maybe add: "than along slantwise trajectories".**
**- Lines 368-370: Part of the phrase is repeated.**

- We have addressed the individual comments in the revised manuscript.